# Urate Levels as a Predictor of the Prevalence and Level of Cardiovascular Risk Factors: An Identificación de La PoBlación Española de Riesgo Cardiovascular y Renal Study

**DOI:** 10.3390/biom14121530

**Published:** 2024-11-29

**Authors:** Paula Antelo-Pais, Miguel Ángel Prieto-Díaz, Rafael M. Micó-Pérez, Vicente Pallarés-Carratalá, Sonsoles Velilla-Zancada, José Polo-García, Alfonso Barquilla-García, Leovigildo Ginel-Mendoza, Antonio Segura-Fragoso, Facundo Vitelli-Storelli, Vicente Martín-Sánchez, Álvaro Hermida-Ameijerias, Sergio Cinza-Sanjurjo

**Affiliations:** 1Santa Comba Health Centre, Health Area of Santiago de Compostela, PC 15840 Santiago de Compostela, Spain; paula.antelo.pais@sergas.es; 2Vallobín-La Florida Health Centre, PC 33012 Oviedo, Spain; maprietodiaz@telefonica.net; 3Fontanars dels Alforins Health Centre, Xàtiva–Ontinyent Department of Health, PC 46635 Valencia, Spain; rafaelmmicoperez@gmail.com; 4Department of Medicine, Jaume I University, PC 12006 Castellón, Spain; 5Joaquin Elizalde Health Centre, PC 26004 Logroño, Spain; svelizan@hotmail.com; 6Casar de Cáceres Health Centre, PC 10190 Cáceres, Spain; jpolog@telefonica.net; 7Trujillo Health Center, PC 10200 Cáceres, Spain; alfonso.barquilla@gmail.com; 8Ciudad Jardín Health Center, PC 04007 Málaga, Spain; lginel@gmail.com; 9Epidemiology Unit, Semergen Research Agency, PC 28009 Madrid, Spain; asegurafr@gmail.com; 10Gene-Environment-Health Interaction Research Group (GIIGAS)/Institute of Biomedicine (IBIOMED), University of León, PC 24004 Leon, Spain; fvits@unileon.es; 11Institute of Biomedicine (IBIOMED), Epidemiology and Public Health Networking Biomedical Research Centre (CIBERESP), University of León, PC 24004 Leon, Spain; vicente.martin@unileon.es; 12Department of Internal Medicine, University Hospital of Santiago de Compostela, PC 15706 Santiago de Compostela, Spain; alvaro.hermida@usc.es; 13Milladoiro Health Centre, Health Area of Santiago de Compostela, PC 15895 Ames, Spain; scinzas@semergen.es; 14Health Research Institute of Santiago de Compostela (IDIS), PC 15706 Santiago de Compostela, Spain; 15Networking Biomedical Research Centre-Cardiovascular Diseases (CIBERCV), PC 28029 Madrid, Spain

**Keywords:** hyperuricemia, cardiovascular risk, primary care

## Abstract

(1) Background: Urate levels lower than the classical cut-off point for defining hyperuricemia can increase cardiovascular risks. The aim of this study is to determine if there is a relationship between different urate levels and classic cardiovascular risk factors (CVRFs). (2) Methods: A cross-sectional study of the inclusion visits of the patients recruited to the IBERICAN study was conducted. The patients were classified into quartiles according to their distribution of urate levels and separated by sex; the three lower points corresponded to normal levels of urate, and the highest quartile was determined according to the classical definition of HU. Multivariate analysis models, adjusted for epidemiological variables, were used to analyze the association of urate levels with CVRFs. (3) Results: The presence of CVRFs was higher across the quartiles of urate, with a continuous increase along the quartiles in both sexes in accordance with body mass index (*p* < 0.01), waist circumference (*p* < 0.01), blood pressure (*p* < 0.01), and LDL cholesterol (*p* < 0.01). The CV risk estimated by SCORE was associated with an increase along the quartiles in women (*p* = 0.02). (4) Conclusions: A progressive increase in the frequency of CVRFs, as well as in their levels, was observed across the quartiles of uricemia, which reflects an increase in the CVRs associated with uricemia.

## 1. Introduction

Atherosclerosis is the pathophysiological basis of cardiovascular disease (CVD). It is a silent process connecting cardiovascular risk factors (CVRFs) to CVD, presenting as irreversible damage. Inflammation plays an important role as a mediator in all the stages of this disease [1].

There is growing evidence that urate may play a pathophysiological role in metabolic disease [2], as well as in the first and second stage of CVD [3]. There are two known mechanisms: inflammation and oxidative stress [4]. However, it also has an antioxidant effect, being responsible for between 50% and 70% of the body’s antioxidant effect [5], acting as an eliminator of waste products such as LDL [6]. Despite the prevalent idea that they may be related to the dysfunction of this antioxidant effect in patients with increased levels of urate [7,8], it is unclear which of these mechanisms is the most important with respect to CVR [8,9].

The accepted classic cut-off values for hyperuricemia (HU) have been defined as urate > 6 mg/dL in women and >7 mg/dL in men based on the urate saturation point and its deposition in the atheromatous plaque; this is specified in the 2018 European Hypertension Guidelines [10]. This definition, based on chemical criteria, has led to confused results between different studies with different cohorts.

Analyses of quartiles differ between studies, with different cut-offs depending on the samples; this may be the reason why the results are different and sometimes contradictory. From the first study, a sub-analysis of the Framingham cohort, which concluded that there was no causal relationship observed [11], to the most recent study, the URRAH study, which concluded that urate is a predictor of total mortality, cardiovascular mortality, and fatal acute myocardial infarction (AMI) [12], there have been many analyses with contradictory results. Some of them have observed an association between elevated levels of urate with CVRFs, such as prehypertension [13], hypertension (HTN) [14], and diabetes mellitus (DM) [15]; others have observed an association with coronary artery disease, stroke [16], and the different subtypes of atrial fibrillation (AF) [17].

Specifically, the analysis of quartiles is infrequent, and the majority of studies only observe a relationship between the highest quartile and subclinical atherosclerosis [18], renal disease [19,20], and cardiovascular mortality [21,22,23], without considering any evidence related to the lower levels. In the same sense that blood pressure and glucose levels increase in intermediate levels that reflect intermediate risk in patients, intermediate levels of urate are potentially associated with moderate risk in patients. Another variable to consider in the HU study with respect to CVR is the influence of each sex. On the one hand, there were differences described in terms of CVRFs [24]; and on the other, there were also differences in the prevalence of AMI [25]. In our opinion, it is necessary to analyze the roles of urate in cardiovascular risk, as there are lower classical levels with respect to CVRFs, as well as how these progress along with urate levels in terms of sex.

The aim of this study is to analyze the association of urate levels with the presence of CVRFs and their respective levels according to sex. Secondly, we analyze the estimated changes in cardiovascular risk along with the different levels of urate.

## 2. Materials and Methods

### 2.1. Study Design

A cross-sectional analysis was performed on patients recruited for the IBERICAN cohort by Spanish primary care physicians (PCPs). This study was approved by the CREC of the Hospital Clínico San Carlos in Madrid on 21 February 2013 (C.P. IBERICAN-C.I. 13/047-E), and the general characteristics of the study have already been published [26].

### 2.2. Patient Selection

The sample of this study consists of 6927 of the 8066 patients of the IBERICAN study from whom urate levels were available. All of them were 18 to 85 years old and were permanently assigned to a PCP [26].

### 2.3. Recorded Variables

Sociodemographic data (sex, age, ethnicity, habitat, level of education, family economic status, current employment status), personal history (HTN, DM, hypercholesterolemia), clinical parameters (weight, height, body mass index (BMI), waist circumference (WC), and systolic and diastolic blood pressure (SBP and DBP)) were collected in the inclusion visit.

HTN was defined as a diagnosis of hypertension, blood pressure levels above 140/90 mmHg, or current use of antihypertensive medication [10]; DM was defined as a diagnosis of diabetes, HbA1c levels above 7%, or use of antidiabetic medication [27]; hypercholesterolemia was defined as a diagnosis of hypercholesterolemia, LDL cholesterol levels above the established thresholds for their risk level, or taking lipid-lowering medication [28].

In addition, the results of blood tests (glucose and cholesterol levels) carried out within the last 6 months were included. Each variable has been defined according to the Clinical Practice Guidelines [26,29].

Finally, CVR stratification of patients was performed according to the SCORE tables for low-risk countries [30].

### 2.4. Statistical Analysis

The qualitative variables were defined as percentages with a 95% confidence interval (95% CI), and the continuous variables were defined as mean with standard deviation (SD) after checking with a Kolmogorov–Smirnov test.

Urate levels were classified by quartiles according to sex with the following cut-off points: Q1 (Men ≤ 4.9 mg/dL; Women ≤ 3.8 mg/dL); Q2 (Men > 4.9–≤ 5.8 mg/dL; Women > 3.8–≤ 4.6 mg/dL); Q3 (Men > 5.8–≤ 6.8 mg/dL; Women > 4.6–≤ 5.5 mg/dL); and Q4 (Men > 6.8 mg/dL; Women > 5.5 mg/dL).

Bivariate analysis was carried out using the Chi-square test for categorical variables, and the ANOVA test was carried out for continuous variables, which was completed with a post hoc analysis (Bonferroni) in cases where statistically significant differences were identified to check which quartiles presented differences.

The means of the continuous quantitative variables (BMI, WC, SBP, DBP, blood glucose, total cholesterol, HDL cholesterol, LDL cholesterol, and triglycerides) were compared with the urate quartiles for each sex using linear regression models, adjusting for age, level of education, physical activity, and adherence to the Mediterranean diet.

For this adjustment, self-reported physical activity was also recorded, and a sedentary lifestyle was defined as less than 30 min of moderate-intensity daily walking for less than 4 days [29]. Adherence to the Mediterranean diet pattern was determined via the Mediterranean Diet Score questionnaire [31].

In all comparisons, the null hypothesis with an alpha error < 0.05 has been rejected. IBM SPSS (Statistical Package for Social Sciences) para Windows (Released 2013, IBM Corp., Armonk, NY, USA), IBM SPSS Statistics for Windows, version 22.0.0.0., IBM Corp., Armonk, NY, USA and STATA 15 (Stata Statistical Software: Release 15, Stata Corp. LLC, College Station, TX, USA) were used for data analysis.

## 3. Results

### 3.1. General Characteristics of the Sample

Mean uricemia levels were 5.88 (1.41) mg/dL in men and 4.71 (1.28) mg/dL in women. Table 1 shows the epidemiological data of patients for the four quartiles; there is an increase in age with each quartile from Q2 (*p* < 0.001), without differences between Q1 and Q2 (*p* = 0.738), and with no other differences in sex or in other variables.

An increase in BMI and WC was observed across the quartiles in both sexes, although in women, it begins at lower levels in the initial quartiles and increases in both parameters of obesity to reach a similar level to men in the highest quartile (*p* < 0.01) (Figure 1 and Appendix A).

SBP (*p* < 0.01) and DBP (*p* < 0.01), adjusted for epidemiological variables, showed a progressive increase across the quartiles, with lower levels in women than in men but with a greater increase in lower quartiles in women (Figure 2 and Appendix A).

Blood glucose showed changes along the quartiles with different progressions in both sexes; there was an increase in women (*p* < 0.01) and a negative association in men (*p* < 0.01), especially in the highest quartile (*p* < 0.01) (Figure 3 and Appendix A).

In both sexes, a similar increase in LDL cholesterol (*p* < 0.01) and triglyceride (*p* < 0.01) levels was observed from Q2 onwards. HDL cholesterol showed a negative relationship in both sexes; however, in women, it was progressive from the lowest quartile; while in men, there was only a significant decrease in the highest quartile (*p* < 0.01) (Figure 4 and Appendix A). We did not observe differences in the number of drugs used to combat hypercholesterolemia (0.84 [0.54] vs. 0.79 [0.54] vs. 0.80 [0.54] vs. 0.82 [0.56], *p* = 0.109), and neither did we observe a difference in statins use (being the most frequently employed drug) (31.6% vs. 32.6% vs. 36.8% vs. 30.8%, *p* = 0.591).

### 3.2. Distribution of Cardiovascular Risk Factors and Estimated Cardiovascular Risk

Overall, a progressive increase in all CVRFs was observed, parallel to the increase in quartiles, except for smoking, which showed a progressive decline but without statistical significance (*p* = 0.565) (Figure 5). In addition, a different predominance of CVRFs was observed at each level of uricemia: dyslipidemia (44.8%) was the most frequent at the lowest level of uricemia, and abdominal obesity (71.6%) was the most frequent at the highest levels.

The CVR estimated by SCORE showed a higher risk in men but without association with urate levels (*p* = 0.11). In women, a progressive increase was observed from Q3 (*p* = 0.02) (Figure 6 and Appendix A).

## 4. Discussion

The results obtained from a sample of 6927 patients recruited in primary care indicate that both the prevalence and the levels of each CVRF are progressively higher across the uricemia levels as classified by quartiles, with the exception of glycemia in men. This increase was observed from the lowest levels of urate, with a progressive evolution along the normal levels to eventually reach the classic HU cut-off points. Although we have described CVR associated with HU in previous studies [32], our results expand on these conclusions because there may be an increased risk in the normal uricemia ranges, taking into account the higher prevalence and higher levels of each CVRF. The analysis by sex confirms this: although most of the parameters analyzed are lower in women than men, their increase with increasing uricemia has a better correlation and is more pronounced in women. This could improve the CVR estimation in women and the identification of patients with higher CVR.

Our analyses have divided urate levels by quartiles, bringing the Q4 points very close to the classic definition of HU [10]. This allowed us to analyze the variation in the CVRFs’ prevalence and values along normal urate levels in the three lower quartiles, as well as the classic HU definition in the highest quartile.

Our results describe an association between uricemia and metabolic disorders, such as both parameters of obesity and blood glucose in women, as well as blood pressure and LDL cholesterol in both sexes. These changes were observed from the lowest urate levels and were especially high in the last quartile. All of them began at lower levels for women, but in some of them, such as BMI, blood glucose, and LDL cholesterol, the highest levels in the last quartile were similar to those in men. Our results could indicate that urate reflects the progression along the cardiovascular continuum, with higher values and prevalence of CVRFs along urate levels, and also identify patients with highest CVR as women with metabolic disorders.

The relationship between uricemia and obesity has already been described by other authors in both sexes, proposing a causal relationship between them [33,34] based on the existing linear relationship [35]. Most studies associate obesity, especially abdominal obesity [36], with HU as defined by the dichotomous criteria of the Guidelines [37]. Our results extend this association by using the two most common criteria for obesity and confirming the correlation from normal urate values. In addition, we reinforce the possible synergistic role of obesity and uricemia in the development of other metabolic disorders such as hypercholesterolemia [38], DM [39], or metabolic syndrome [40], with a particular association in women with a higher CVR [41,42]. In the analyses of continuous variables related to cholesterol levels, we observed an increase in total cholesterol, LDL cholesterol, and triglycerides and a reduction in HDL cholesterol across urate quartiles. These results are consistent with the PAMELA study. Maloberti et al. observed an association between the lipid profile and adiposity indexes and HU [43], which could explain the effect of oxidative stress in the lipid profile and the related CVR [44].

Previous studies, such as the TCLSIH (Tianjin Chronic Low-grade Systemic Inflammation and Health) [45] and the STANISLAS (Suivi Temporaire Annuel Non-Invasif de la Santé des Lorrains Assurés Sociaux) studies [13,46], have described a higher incidence of HTN and prehypertension in patients with previous HU. A very interesting recent work observed a relationship between uricemia and non-dipper blood pressure with an increase in CVR [47]. Our results extend this association, describing a progressive increase in blood pressure levels—both systolic and diastolic—in both sexes from very low urate values, which is maintained across the quartiles. Maybe the association between uricemia and HTN could be related to treatment with thiazides, but previous analyses in our sample showed that it was independent [32].

As discussed earlier, these results show that the progressive increase in normal urate levels is the reflection of the first phases of the cardiovascular continuum [48], i.e., when the patient only has CVRFs, and HU levels correspond to the advanced stages of this continuum, with a high prevalence of CVD [32]. Similar results were described in the URRAH study; Maloberti et al. proposed a lower level for defining HU based on their finding of an association between target organ damage (TOD) and lower urate levels [49].

Our results also showed differences between sexes. We found a high impact of urate levels in women, with a high correlation with some CVRFs, as well as a higher correlation between CVR and urate levels. There are no studies that analyze this association, but some works describe a higher association between urate levels and CVD in women than in men, such as the URRAH study [25] or Sun et al. [50].

### Strengths and Limitations

Our results are statistically robust and consistent with the existing literature on this topic. The sample size and the recruitment of patients in primary care gives our work sufficient statistical power to answer the research question posed, with good representativeness of the sample and generalization of the results, which would allow for the better applicability of the results to a larger number of patients. The sex-disaggregated analysis confirms differences in the association of urate with cardiovascular risk factors, sometimes differing from those described in the general sample, and thus enabling us to draw more accurate conclusions.

As said before, our overall results are consistent with the published literature, but we have contributed important knowledge by describing a progressive increase in all of the analyzed cardiovascular risk factors across urate values which are usually considered normal. This could reflect an increased cardiovascular risk throughout the distribution of these values.

However, our study also has limitations. Some have been described in previous publications [48,51], but others should be mentioned here due to their relationship with this work. On the one hand, the analyses were not carried out in a centralized laboratory, and the values used for the analysis were those provided by the reference laboratories of each health center participating in the study; in any case, however, they are all laboratories of the Spanish National Health System, which reflect the usual clinical practice. We should also point out that patients receiving urate-lowering treatments have not been excluded. This would classify patients who actually have HU as “healthy”, who may have a higher incidence of events than actually healthy patients. Thus, the differences we describe could be greater, which would confirm our conclusions. In spite of these limitations, and taking into account that the results have been obtained in a large sample of patients who were followed up by their PCP, it should be noted that our findings show the association of uricemia with CVRFs from very low urate levels, i.e., lower than those considered HU. 

On the other hand, some might miss an analysis with other diseases, such as gout or metabolic syndrome. But metabolic syndrome is defined by the coexistence of risk factors that we have included in our results; if our results showed an increase along with urate levels, a progressive increase in all of them can be expected [49]. The diagnosis of gout, although associated with cardiovascular risk, is more a consequence of usually elevated urate levels than an association with cardiovascular risk or risk factors. An adjustment of our model to include gout would only neutralize the effect of elevated urate levels as it is at this level that the highest number of gout episodes occur [21].

This relationship broadens the concept of CVR associated with HU, in which there is a higher incidence of CV events [51]. It also highlights the possibility that urate levels below HU diagnostic values should be included as an estimation variable for CVR. This would give uricemia a role as a facilitator or an enhancer of CVR in patients with CVRFs. In any case, this prognostic association should be confirmed in prospective studies that analyze their incidence.

## 5. Conclusions

We can conclude that there is an association between the most classic CVRFs and uricemia, beginning from very low urate levels, and that this association is progressive and positive across all levels of uricemia. This has been seen in both sexes, although it could have a greater impact on women, especially at higher urate values.

In the CVR study, our results show that urate levels can reflect the stage of the cardiovascular continuum where the patient is, i.e., lower levels reflect its beginning, with presence of CVRFs, which increase progressively to higher levels, associated with subclinical target organ damage, and, finally, to HU levels that lead to CVD.

Longitudinal studies are needed in order to analyze the incidence of cardiovascular and renal events across the range of urate values, providing more information in this regard.

## Figures and Tables

**Figure 1 biomolecules-14-01530-f001:**
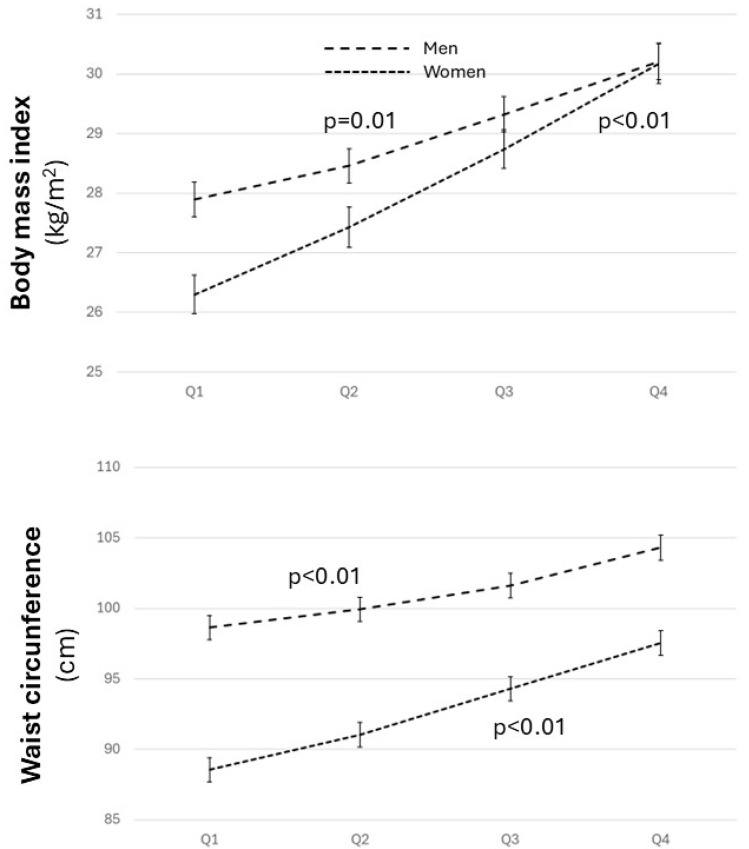
Body mass index and waist circumference across the quartiles for both sexes. ANOVA for the relationship between body mass index and waist circumference in both sexes, adjusted for age, level of education, physical activity, and adherence to the Mediterranean diet.

**Figure 2 biomolecules-14-01530-f002:**
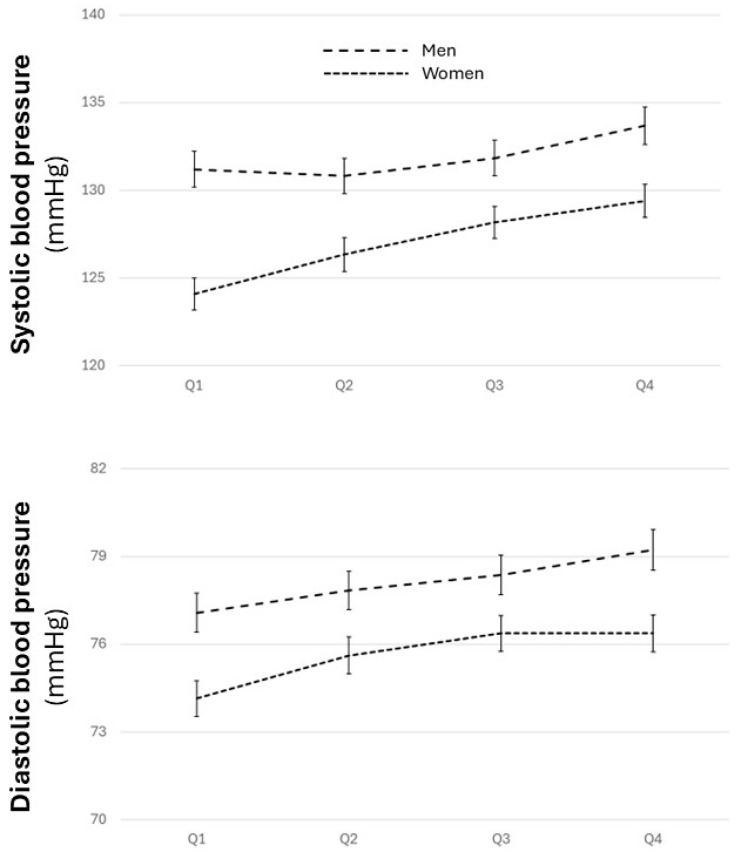
Systolic and diastolic blood pressure across the quartiles for both sexes. ANOVA for the relationship between systolic and diastolic blood pressure in both sexes, adjusted for age, level of education, physical activity, and adherence to the Mediterranean diet.

**Figure 3 biomolecules-14-01530-f003:**
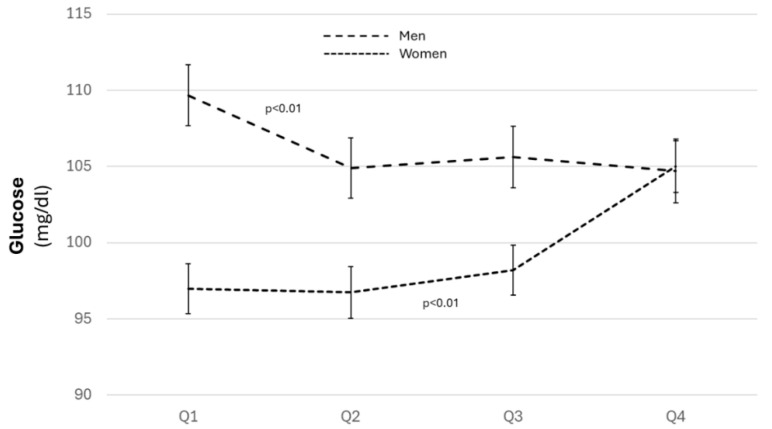
Levels of blood glucose across the quartiles for both sexes. ANOVA for the relationship between blood glucose in both sexes, adjusted for age, level of education, physical activity, and adherence to the Mediterranean diet.

**Figure 4 biomolecules-14-01530-f004:**
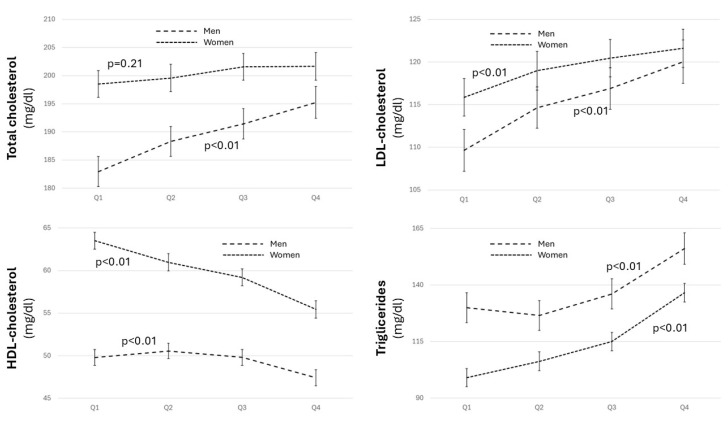
Lipid profile levels across the quartiles for both sexes. ANOVA for the lipid profile relationship in both sexes, adjusted for age, level of education, physical activity, and adherence to the Mediterranean diet.

**Figure 5 biomolecules-14-01530-f005:**
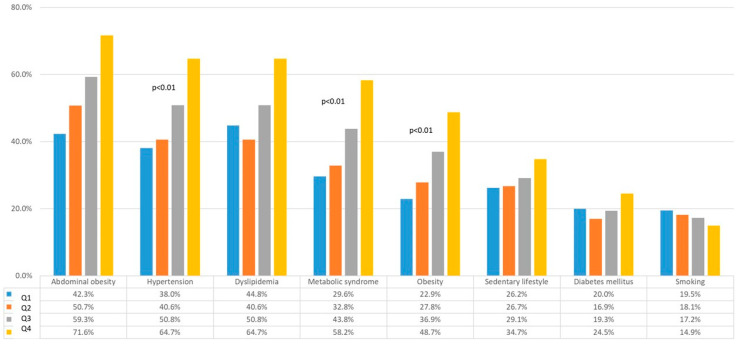
Prevalence of cardiovascular risk factors in each quartile.

**Figure 6 biomolecules-14-01530-f006:**
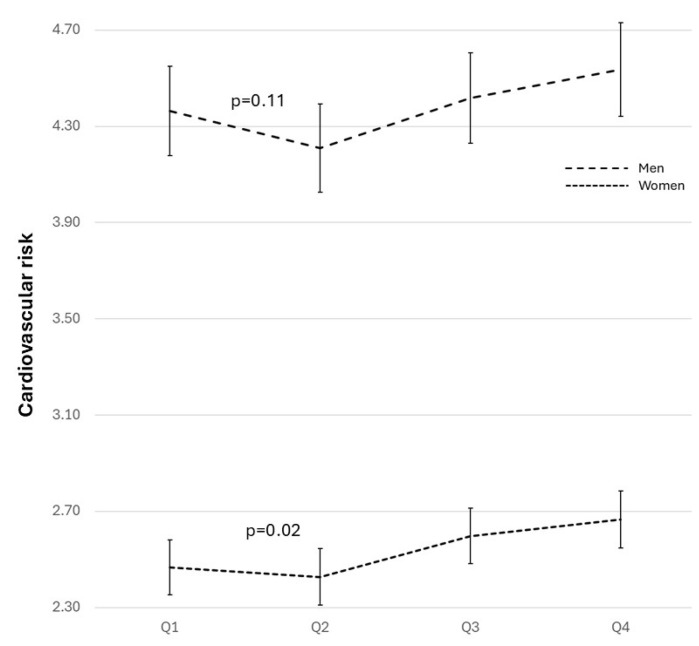
SCORE-estimated cardiovascular risk for both sexes according to the four levels of uricemia. ANOVA for the SCORE risk level relationship in both sexes, adjusted for age, level of education, physical activity, and adherence to the Mediterranean diet.

**Table 1 biomolecules-14-01530-t001:** Sociodemographic characteristics of patients in each quartile.

	Q1		Q2		Q3		Q4		*p*
	1609		1821		1697		1800		
	N	Mean [SD]	N	Mean [SD]	N	Mean [SD]	N	Mean [SD]	
**Age (years)**	1609	55.91 [14.94]	1821	56.91 [15.06]	1697	59.55 [14.14]	1800	62.15 [13.55]	<0.001
	**N**	**%**	**N**	**%**	**N**	**%**	**N**	**%**	
**Women**	873	54.3%	993	54.5%	878	51.7%	995	55.3%	0.136
**Ethnicity**									
White	1545	96.0%	1747	95.9%	1641	96.7%	1745	96.9%	0.656
Black	4	0.2%	7	0.4%	7	0.4%	12	0.7%	
Latin American	49	3.0%	55	3.0%	41	2.4%	34	1.9%	
Asian	2	0.1%	1	0.1%	1	0.1%	3	0.2%	
Berber	9	0.6%	11	0.6%	7	0.4%	6	0.3%	
**Habitat**									
Urban	947	58.9%	1056	58.0%	999	58.9%	1059	58.8%	0.986
Semi-urban	342	21.3%	398	21.9%	353	20.8%	381	21.2%	
Rural	319	19.8%	367	20.2%	344	20.3%	360	20.0%	
**Education**									
Unschooled	116	7.2%	126	6.9%	160	9.4%	225	12.5%	0.124
Primary education	863	53.6%	1014	55.7%	931	54.9%	1006	55.9%	
Higher education	377	23.4%	430	23.6%	376	22.2%	378	21.0%	
University education	253	15.7%	251	13.8%	230	13.6%	191	10.6%	
**Employment**									
Jobholder	739	46.0%	810	44.6%	718	42.5%	647	36.0%	0.138
Unemployed	141	8.8%	160	8.8%	143	8.5%	125	7.0%	
Retired	493	30.7%	609	33.6%	617	36.5%	766	42.7%	
Student	23	1.4%	29	1.6%	15	.9%	9	.5%	
Housekeeping	209	13.0%	207	11.4%	197	11.7%	248	13.8%	
**Income**									
Annual income lower than 18,000 EUR	664	41.3%	744	40.9%	719	42.4%	793	44.1%	0.935
Annual income between 18,000 EUR and 100,000 EUR	927	57.6%	1057	58.0%	953	56.2%	983	54.6%	
Annual income higher than 100,000 EUR	18	1.1%	20	1.1%	25	1.5%	24	1.3%	

Quantitative variables are presented as mean [SD]; qualitative variables are presented as %.

## Data Availability

The authors may provide the data used for the development of this article upon a justified request addressed to the corresponding author.

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
