# Peer review of "Urate Levels as a Predictor of the Prevalence and Level of Cardiovascular Risk Factors: An Identificación de La PoBlación Española de Riesgo Cardiovascular y Renal Study"

_biomolecules, 2024, doi:10.3390/biom14121530_

Round 1

Reviewer 1 Report

Comments and Suggestions for Authors

Cross-sectional study of patient in the IBERICAN study. Quartiles of urate associated with cardiovascular (CV) risk factors for 6,300 patients.

The graphs nicely present the progressive increased CVRF across urate levels.

These are nice findings. I think the paper could be strengthened with additional analyses.

MAJOR comments:

The correlation of CVRF and urate is interesting, but it begs the question about more complex modeling. Urate quartile ought to be controlled for sex, age, renal function, meds, diagnosis of gout, obesity (for example). I don't find the current multivariable analysis sufficient (age, education, activity, diet). The paper is lacking a conceptual model to organize the analysis, and I think this is needed. Simply presenting correlations (in nice figures), I feel is insufficient.

Consider defining primary (e.g. SCORE) and secondary outcomes (e.g. unique CVRFs).  (Manuscript needs primary and secondary outcomes based on stated hypotheses/aims - conceptual model)

The finding of progression of CVRF with increasing urate quartile is pursuasive that URATE not GOUT or HYPERURICEMIA is potential driving factor. (Of course it could just be common predisposing metabolic syndrome genetic factor driving CVRF and URATE rather than URATE being causative job CVRF). More discussion on this in limitation section is needed.

I'm not persuaded by the final paragraph of Conclusion. I think this overstates the value of the findings.

MINOR comments:

Urate is preferred term over uric acid. (Refer to G-CAN nomenclature paper)

Line 63, yes SU > 6 and > 7 OK for HU. I don't follow "and its deposition in the atheromatous plaque". Reference is good review, but it does not tie in with this statement. I suggest simply delete.

Formatting for Table 1 needs fixing. It is miserably hard to read in current structure.

In contract, I found Figures very east to read.. For BMI, I would like to see Quartile*SEX interaction term (to describe if there is statistical meaningful difference int he slopes). Same for TOT_CHOL and HDL. 

Figure 5 is very clear. How is SCORE so poorly correlated with quantiles where all the factors going into SCORE strongly correlated?

Author Response

Comments and Suggestions for Authors

Cross-sectional study of patient in the IBERICAN study. Quartiles of urate associated with cardiovascular (CV) risk factors for 6,300 patients. The graphs nicely present the progressive increased CVRF across urate levels. These are nice findings. I think the paper could be strengthened with additional analyses.

MAJOR comments:

The correlation of CVRF and urate is interesting, but it begs the question about more complex modeling. Urate quartile ought to be controlled for sex, age, renal function, meds, diagnosis of gout, obesity (for example). I don't find the current multivariable analysis sufficient (age, education, activity, diet). The paper is lacking a conceptual model to organize the analysis, and I think this is needed. Simply presenting correlations (in nice figures), I feel is insufficient.

We appreciate the reviewer's effort, and the suggestions provided, which we are confident will improve the manuscript. We will attempt to address them with the appropriate modifications.

The analysis shown adjusts the results for the covariates that have shown an association with urate levels in previous works. Although we agree with the adjustment proposed by the reviewer, our results are already presented separately by sex, so an adjustment for sex is not necessary, and we have used age in the provided model. On the other hand, the diagnosis of gout, although associated with cardiovascular risk, is more a consequence of urate levels, usually elevated levels, than an association with cardiovascular risk or risk factors. An adjustment of our model including gout would only neutralize the effect of elevated urate levels, as it is at this level that the highest number of gout episodes occur (Stack AG, Hanley A, Casserly LF, Cronin CJ, Abdalla AA, Kiernan TJ, et al. Independent and conjoint associations of gout and hyperuricaemia with total and cardiovascular mortality. QJM: An International Journal of Medicine. 2013;106(7):647–58).

Obesity, measured in our work by BMI and WC, is one of the outcomes analyzed, so it is also not appropriate to adjust for this variable, as the analysis objectives should not be part of the adjustment model.

Finally, both renal function and some medications, especially thiazides, have been observed to be associated with urate levels in our cohort as well (Antelo-Pais, P.; Prieto-Díaz, M.Á.; Micó-Pérez, R.M.; Pallarés-Carratalá, V.; Velilla-Zancada, S.; Polo-García, J.; Barquilla-García, A.; Ginel-Mendoza, L.; Segura-Fragoso, A.; Vitelli-Storelli, F.; et al. Prevalence of Hyperuricemia and Its Association with Cardiovascular Risk Factors and Subclinical Target Organ Damage. J Clin Med 2022, 12, 50, doi:10.3390/jcm12010050). In both cases, although the association has been significant and analyzed in previous works as referenced, it has been independent of the factors we have included in the model.

Our objective in the present manuscript has been to analyze the association between urate levels and factors that may be precursors of renal and cardiovascular events, such as epidemiological variables, lifestyle habits, or cardiovascular risk factors, allowing us to complement previous results, such as the referenced one, and to establish a risk model to analyze the incidence of events. The presented results reflect that the analyzed factors show a progressive increase across the defined quartiles, with a parallel increase in cardiovascular risk in patients as shown in Figure 5. Our results expand the evidence described by other authors in specific patient groups such as hypertension (Verdecchia P, Schillaci G, Reboldi G, Santeusanio F, Porcellati C, Brunetti P. Relation Between Serum Uric Acid and Risk of Cardiovascular Disease in Essential Hypertension. Hypertension. 2000;36(6):1072–8), diabetes mellitus (Jenkins AJ, Braffett BH, Basu A, Bebu I, Dagogo-Jack S, Orchard TJ, et al. Serum urate and cardiovascular events in the DCCT/EDIC study. Scientific reports. 2021;11(1):14182), obese individuals (Kahaer M, Zhang B, Chen W, Liang M, He Y, Chen M, et al. Triglyceride Glucose Index Is More Closely Related to Hyperuricemia Than Obesity Indices in the Medical Checkup Population in Xinjiang, China. Frontiers in endocrinology. 2022), or even in patients with subclinical lesions (Drivelegka P, Forsblad-D’elia H, Angerås O, Bergström G, Schmidt C, Jacobsson LTH, et al. Association between serum level of urate and subclinical atherosclerosis: results from the SCAPIS Pilot. Arthritis research & therapy. 2020;22(1)), and the risk of recurrent events in patients with previous cardiovascular disease (Lim SS, Yang YL, Chen SC, Wu CH, Huang SS, Chan WL, et al. Association of variability in uric acid and future clinical outcomes of patient with coronary artery disease undergoing percutaneous coronary intervention. Atherosclerosis. 2020;297(201):40–6).

We have included some of previous comments in the discussion section.

Although the reviewer considers our analysis simple, from our point of view, it provides interesting results that allow us to propose a working hypothesis for future analyses, so that urate may increase along the cardiovascular continuum, parallel to the patient's cardiovascular risk, reflecting an increase in the values of each risk factor and even, pending our analysis, a progressive increase in the incidence of renal and cardiovascular events. All the references provided in this response represent works that have analyzed urate by quartiles, suspecting an increase in risk at values considered normal with a single cut-off point, although different by sex. Our work confirms this increase in risk factors, and when the follow-up of patients is longer, with greater statistical power, we will analyze whether there are differences in the incidence of events.

Consider defining primary (e.g. SCORE) and secondary outcomes (e.g. unique CVRFs).  (Manuscript needs primary and secondary outcomes based on stated hypotheses/aims - conceptual model)

In accordance with the reviewer's suggestions, we will establish a primary and a secondary objective. However, if permitted, we prefer to establish the association analyses with risk factors as the primary objective and the SCORE as the secondary objective. It is important to note that the SCORE used is a risk equation; we have not analyzed the incidence of events, which of course would be the primary objective. An increase in risk factors is expected to elevate the risk. From our perspective, the increase in risk factors with urate levels is much more significant, and as a secondary consequence, this results in a higher estimated risk, which should be verified by a higher incidence of events.

We have redrafted the objectives to include this secondary objective.

The finding of progression of CVRF with increasing urate quartile is persuasive that URATE not GOUT or HYPERURICEMIA is potential driving factor. (Of course it could just be common predisposing metabolic syndrome genetic factor driving CVRF and URATE rather than URATE being causative job CVRF). More discussion on this in the limitation section is needed.

Metabolic syndrome has not been analyzed as it is defined by the coexistence of risk factors that we have included in our results. An aggregated analysis of a single disease would not provide new insights since we have described a progressive increase in all of them. Therefore, it is expected that there would be an increase in the prevalence of metabolic syndrome with urate levels, as well as in the values of each defining element (Mena-Sánchez G, Babio N, Becerra-Tomás N, Martínez-González MÁ, Díaz-López A, Corella D, et al. Association between dairy product consumption and hyperuricemia in an elderly population with metabolic syndrome. Nutrition, Metabolism and Cardiovascular Diseases. 2020;30(2):214–22).

As previously mentioned, hyperuricemia has been analyzed in prior studies. The present work aims to extend the analysis to determine if there is a progressive increase in risk factors from values considered normal, which could expand the study of urate’s role in cardiovascular risk. Additionally, gout is associated with elevated urate levels and therefore would not provide much more information beyond what is already indicated by this biomarker.

We have included some of previous comments in the discussion section as limitations.

I'm not persuaded by the final paragraph of Conclusion. I think this overstates the value of the findings.

According with your comment, we have modified the last paragraph of the conclusions: “It is necessary to develop longitudinal studies that analyze the incidence of cardiovascular and renal events across the range of urate values, providing more information in this regard”.

MINOR comments:

Urate is preferred term over uric acid. (Refer to G-CAN nomenclature paper)

We have changed the word “uric acid” for “urate” along the document.

Line 63, yes SU > 6 and > 7 OK for HU. I don't follow "and its deposition in the atheromatous plaque". Reference is good review, but it does not tie in with this statement. I suggest simply delete.

We have deleted the reference.

Formatting for Table 1 needs fixing. It is miserably hard to read in current structure.

We have deleted the columns with the post-hoc analyses and include the p values in table food. Maybe the actual design is more friendly to review the data.

In contrast, I found Figures very east to read. For BMI, I would like to see Quartile*SEX interaction term (to describe if there is statistical meaningful difference int he slopes). Same for TOT_CHOL and HDL. 

We have included in the supplementary material a table with the numeric parameters and the p value. This p value is compared with the other 3 numeric parameters in the other quartiles. This exposition is more correct, and we think that is easier to understand the figures, where we only included one p-value because all of them are the same.

Figure 5 is very clear. How is SCORE so poorly correlated with quantiles where all the factors going into SCORE strongly correlated?

We agree with the reviewer's comment, and our expectation was the same with a greater increase in estimated risk with urate levels. However, the variable that conditions the greatest increase in risk is age, and in a model adjusted for this variable, it is more challenging to observe differences. On the other hand, while the observed differences are statistically significant, it is difficult to establish the clinical impact in terms of the incidence of cardiovascular events. We have used the SCORE, which estimates fatal events, so the possibility of seeing an increased risk is more limited than with scales that analyze both fatal and non-fatal events, like the current SCORE2.

From our point of view, in light of the presented results, we understand that the increase is sufficient considering the scale used and the increments observed in each risk factor. Without a doubt, it is necessary to longitudinally analyze the incidence of cardiovascular events to better understand if there is a gradation of cardiovascular risk with the progressive increase of urate levels.

Reviewer 2 Report

Comments and Suggestions for Authors

The study is interesting to read for both professionals and the public. Some improvements are necessary to increase the impact:

1. The introduction is difficult to understand (some re-writing is needed because the quality of the English language is low).

2. Table 1 needs to be redesigned because it is difficult to read to values and understand the written idea.

3. There are too many figures in the manuscript and the reading gets disturbed. Consider moving some of the figures (e.g. Fig 1-4) to the supplementary files. Also, the results could be summarized in table form and inserted into the manuscript.

4. page 7, lines 171 - 173: HDL-cholesterol and glucose (for men) are also negatively correlated with CVRF.

5. What is the originality of this study? The strengths of the study are not well described or pointed out (I like how the authors have described the limitations and pointed out what the implications could lead up to ).

6. The inclusion of the patients under treatment for high uric acid could give misleading results. You should include a paragraph describing the error that this limitation brings to your study in the "Results" section.

Comments on the Quality of English Language

The English needs some improvements so that the meaning of the sentences is enhanced (I had some small difficulties in understanding the introduction part).

Author Response

REVIEWER 2 comments

Comments and Suggestions for Authors

  1. Authors should use mean with standard deviation in case of normally distributed data or median with lower/upper quartile in other instances for interval data. Please change this in the tables and text.

We want to thank the revisor for the comment. It was a mistake, as we indicated in the methods section, we have used mean and standard deviation. In our wide sample, the variables achieve a normal distribution (Kolmogorov-Smirnov) and we have used parametric statistics.

  1. What test was used to check the data distribution?

As we have indicated previously, we have used Kolmogorov-Smirnov. We have included in the methods section this statistic.

  1. What was the rationale for using quartiles instead of cut-offs well established for hyperuricemia in men and women (6.0 and 6.8 mg/dl)?

The classical cut-off points assume that there is only risk starting from the values that define hyperuricemia. However, these values are described because they reflect the precipitation point of urate, which would correspond to its deposition in the atheroma plaque, associating it with cardiovascular risk. Despite this physiopathologically easy-to-understand approach, the reality is that many authors have described the possibility of increased cardiovascular risk starting from lower values, as confirmed by our results, with a progressive increase in the prevalence and levels of cardiovascular risk factors.

Our results could help rethink the true role of urate, so that possibly it is not a cardiovascular risk factor, but it does reflect the patient's level of risk, indicating an increase in risk factors at lower levels and an increase in cardiovascular events at higher levels. Logically, more studies are needed in this regard, but we believe that this would help to better understand the pathophysiological pathway that explains the role of urate in cardiovascular risk.

  1. Please remove multiple comparisons in Table 1 and discuss them in the text.

We have deleted the columns with the post-hoc analyses and include the p values in the text. Maybe the actual design is more friendly to review the data.

  1. What kind of post-hoc test was used in ANOVA analysis? Please add a description in the statistical analysis.

We have used Bonferroni. We have included in the methods section.

  1. Please add the table as supplementary material with data and ANOVA results used to present figures 1-4 and 6.

We have included in supplementary materials the complete analyses of linear regression (not ANOVA) used in the figures 1-4 and 6.

  1. Please indicate exactly what p-values stand for. What comparison is presented with the p-value?

The p values indicate the comparison between the quartiles. Maybe it is easier to understand in the tables included in supplementary material. The p-values shown in the table of supplementary material corresponding to comparisons between values in each quartile with the other quartiles. In the figure only included one p-value because all of them are the same.

  1. Where are the results of the linear models?

They are included in supplementary material.

  1. Please add the results of the comparison in Table 5.

Maybe the revisor refers to Figure 5. We have changed the figure including the table with the data.

  1. Please add the results of ordered logistic regression with factors influencing the SUA quartiles.

We have included them in the supplementary material.

  1. As the cardiovascular risk and metabolic syndrome-related to SUA are strongly related to hyperlipidemia and hypercholesterolemia, please add data showing the percentage of subjects treated with such medications.

We have analysed the number of drugs used to hypercholesterolemia, without differences in the number (0.84 [0.54] vs 0.79 [0.54] vs 0.80 [0.54] vs 0.82 [0.56], p=0.109). Also, we have analysed the use of statins, as the most used drug, with a light increase but without statistical differences (31.6% vs 32.6% vs 36.8% vs 30.8%, p=0.591).

We have included these results in the manuscript according to your comments.

  1. Please add in the methods definition of states presented in Figure 5.

We have defined in the methods section the definitions of HTN as a patient diagnosed with the condition, with levels above 140/90 mmHg or taking antihypertensive medication [26], DM as a patient diagnosed with the condition, with HbA1c levels above 7%, or taking antidiabetic medication [27]; hypercholesterolemia as a patient diagnosed with the condition, with LDL-cholesterol levels above the established thresholds for their risk level or taking lipid-lowering medication.

Reviewer 3 Report

Comments and Suggestions for Authors

1. Authors should use mean with standard deviation in case of normally distributed data or median with lower/upper quartile in other instances for interval data. Please change this in the tables and text.

2. What test was used to check the data distribution?

3. What was the rationale for using quartiles instead of cut-offs well established for hyperuricemia in men and women (6.0 and 6.8 mg/dl)?

4. Please remove multiple comparisons in Table 1 and discuss them in the text.

5. What kind of post-hoc test was used in ANOVA analysis? Please add a description in the statistical analysis.

6. Please add the table as supplementary material with data and ANOVA results used to present figures 1-4 and 6.

7. Please indicate exactly what p-values stand for. What comparison is presented with the p-value?

8. Where are the results of the linear models?

9. Please add the results of the comparison in Table 5.

10. Please add the results of ordered logistic regression with factors influencing the SUA quartiles.

11. As the cardiovascular risk and metabolic syndrome-related to SUA are strongly related to hyperlipidemia and hypercholesterolemia, please add data showing the percentage of subjects treated with such medications.

12. Please add in the methods definition of states presented in Figure 5.

Author Response

REVIEWER-3 comments

  1. Authors should use mean with standard deviation in case of normally distributed data or median with lower/upper quartile in other instances for interval data. Please change this in the tables and text.

The Table 1 had a mistake in the titles, the correct term was mean and not median. In the actual version it has been corrected.

  1. What test was used to check the data distribution?

We have used Kolmogorov-Smirnov. We have included in the methods section this statistic.

  1. What was the rationale for using quartiles instead of cut-offs well established for hyperuricemia in men and women (6.0 and 6.8 mg/dl)?

The classical cut-off points assume that there is only risk starting from the values that define hyperuricemia. However, these values are described because they reflect the precipitation point of urate, which would correspond to its deposition in the atheroma plaque, associating it with cardiovascular risk. Despite this physiopathologically easy-to-understand approach, the reality is that many authors have described the possibility of increased cardiovascular risk starting from lower values, as confirmed by our results, with a progressive increase in the prevalence and levels of cardiovascular risk factors.

Our results could help rethink the true role of urate, so that possibly it is not a cardiovascular risk factor, but it does reflect the patient's level of risk, indicating an increase in risk factors at lower levels and an increase in cardiovascular events at higher levels. Logically, more studies are needed in this regard, but we believe that this would help to better understand the pathophysiological pathway that explains the role of urate in cardiovascular risk.

  1. Please remove multiple comparisons in Table 1 and discuss them in the text.

We have deleted the columns with the post-hoc analyses and include the p values in table food. Maybe the actual design is more friendly to review the data.

  1. What kind of post-hoc test was used in ANOVA analysis? Please add a description in the statistical analysis.

We have used Bonferroni. We have included in the methods section.

  1. Please add the table as supplementary material with data and ANOVA results used to present figures 1-4 and 6.

We have included in the supplementary material the numeric data used to design the figures 1-4 and 6.

  1. Please indicate exactly what p-values stand for. What comparison is presented with the p-value?

This p value is compared with the other 3 numeric parameters in the other quartiles. This exposition is more correct, and we think that is easier to understand the figures, where we only included one p-value because all of them are the same.

Maybe in the supplementary material is more correct the exposition.

  1. Where are the results of the linear models?

We have included in the supplementary material the numeric data used to design the figures 1-4 and 6.

  1. Please add the results of the comparison in Table 5.

Maybe the revisor refers to Figure 5. We have changed the figure including the table with the data.

  1. Please add the results of ordered logistic regression with factors influencing the SUA quartiles.

Really, we did a linear regression. We have included them in the supplementary material as a table.

  1. As the cardiovascular risk and metabolic syndrome-related to SUA are strongly related to hyperlipidemia and hypercholesterolemia, please add data showing the percentage of subjects treated with such medications.

We have analysed the number of drugs used to hypercholesterolemia, without differences in the number (0.84 [0.54] vs 0.79 [0.54] vs 0.80 [0.54] vs 0.82 [0.56], p=0.109). Also, we have analysed the use of statins, as the most used drug, with a light increase but without statistical differences (31.6% vs 32.6% vs 36.8% vs 30.8%, p=0.591).

We have included these results in the manuscript according to your comments.

  1. Please add in the methods definition of states presented in Figure 5.

We have defined in the methods section the definitions of HTN as a patient diagnosed with the condition, with levels above 140/90 mmHg or taking antihypertensive medication [26], DM as a patient diagnosed with the condition, with HbA1c levels above 7%, or taking antidiabetic medication [27]; hypercholesterolemia as a patient diagnosed with the condition, with LDL-cholesterol levels above the established thresholds for their risk level or taking lipid-lowering medication.

Round 2

Reviewer 2 Report

Comments and Suggestions for Authors

Although the responses the authors gave were not to my comments, I have read the improvements to the paper and agree with them. I still maintain the comments to: 1) re-organize the introduction and introduce a paragraph focusing on the originality of your study and 2) expand the strengths of your study (four lines is insufficient for a 17 pages paper).

Author Response

Although the responses the authors gave were not to my comments, I have read the improvements to the paper and agree with them. I still maintain the comments to:

1) re-organize the introduction and introduce a paragraph focusing on the originality of your study

We would like to thank the reviewer for their time and effort in reviewing the article and helping to improve it to meet their quality standards. Below, we proceed to explain the changes made to the manuscript to accommodate their comments.

We have thoroughly revised the introduction, highlighting the novel aspects our work contributes in relation to the available evidence regarding hyperuricemia analysis and its analysis by quartiles, thus framing the relevance of our work.

The most notable ideas of our manuscript are that the conclusions were obtained from a large sample (6,927 patients) recruited in primary care, for whom we know both cardiovascular risk factors, their values and treatments, in addition to epidemiological factors, allowing us to control all the variables and analyze their impact on cardiovascular risk analysis. This analysis was conducted considering the different levels of urate, classifying them into three levels, with Q4 coinciding with the defining values of hyperuricemia. This has allowed us to analyze whether there is a progression of risk factors in their prevalence but also in their numerical values, which would reveal that urate also has intermediate values that could reflect the patient's cardiovascular risk.

2) expand the strengths of your study (four lines is insufficient for a 17 pages paper).

We have extensively revised the "Strengths and Limitations" section to incorporate the strengths of the study and the results provided, which, along with the previously described limitations, we believe can help the reader/researcher of your journal to better understand and contextualize the presented results.

We mainly identified the most important strengths as the size and representativeness of the sample, the sex-disaggregated analysis, and the analysis of normal values by quartiles, which allowed us to analyze the progression of other cardiovascular risk factors in relation to urate.

We hope that the new version aligns with your vision and meets the quality standards expected in your journal.

Reviewer 3 Report

Comments and Suggestions for Authors

The paper may be published in the current form.

Author Response

The paper may be published in the current form.

Thank you for the previous comments of the reviewer and this new positive answer about the final version.